# SCALEWEAVER: WEAVING EFFICIENT CONTROLLABLE T2I GENERATION WITH MULTI-SCALE REFERENCE ATTENTION

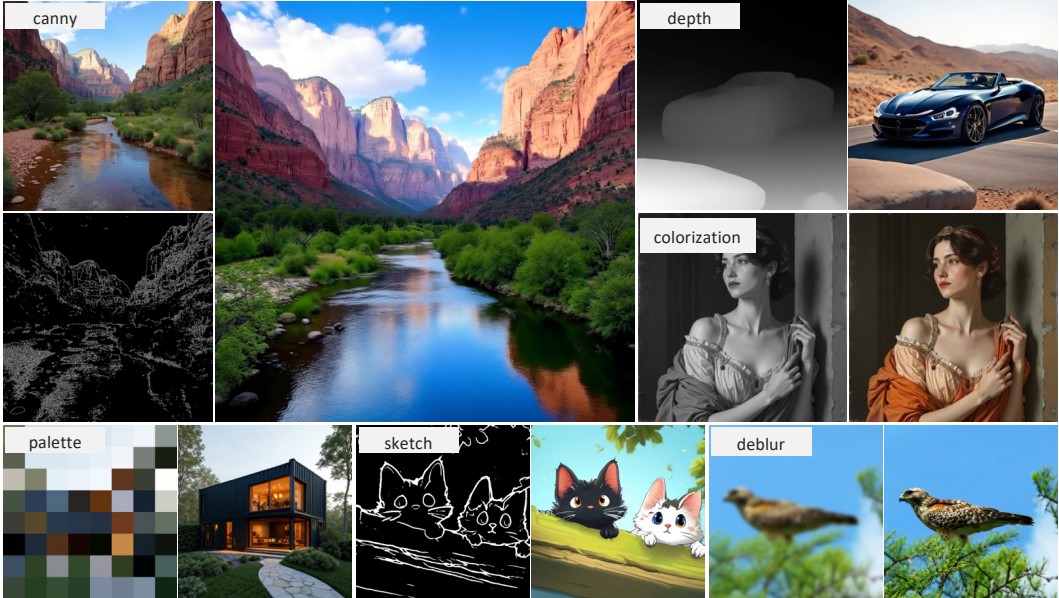

Figure 1: **Images generated by ScaleWeaver.** Our ScaleWeaver enables efficient and precise controllable text-to-image generation based on the visual autoregressive model, supporting diverse condition signals and producing high-fidelity images with strong text alignment and controllability.

## ABSTRACT

Text-to-image generation with visual autoregressive (VAR) models has recently achieved impressive advances in generation fidelity and inference efficiency. While control mechanisms have been explored for diffusion models, enabling precise and flexible control within VAR paradigm remains underexplored. To bridge this critical gap, in this paper, we introduce ScaleWeaver, a novel framework designed to achieve high-fidelity, controllable generation upon advanced VAR models through parameter-efficient fine-tuning. The core module in ScaleWeaver is the improved MMDiT block with the proposed Reference Attention module, which efficiently and effectively incorporates conditional information. Different from MM Attention, the proposed Reference Attention module discards the unnecessary attention from image→condition, reducing computational cost while stabilizing control injection. Besides, it strategically emphasizes parameter reuse, leveraging the capability of the VAR backbone itself with a few introduced parameters to process control information, and equipping a zero-initialized linear projection to ensure that control signals are incorporated effectively without disrupting the generative capability of the base model. Extensive experiments show that ScaleWeaver delivers high-quality generation and precise control while attaining superior efficiency over diffusion-based methods, making ScaleWeaver a practical and effective solution for controllable text-to-image generation within the visual autoregressive paradigm. Code and models will be released.

# 1 INTRODUCITON

Generative models have achieved remarkable progress in recent years, with two dominant paradigms emerging: diffusion models and autoregressive (AR) models. The rise of diffusion-based text-to-image (T2I) models has brought unprecedented fidelity and diversity to generative modeling (Rombach et al., 2022; Podell et al., 2024; Chen et al., 2024; Esser et al., 2024; Wu et al., 2025). In particular, controllable image generation with specified spatial conditions such as edges, depth, or segmentation maps has been extensively explored in diffusion models, giving rise to methods that enable precise and flexible control (Zhang et al., 2023; Mou et al., 2024). These conditional generation approaches have shown great practical value, enabling applications such as guided content creation, image editing, and domain-specific generation in a wide range of scenarios.

In parallel, Autoregressive (AR) models leverage the scaling properties and causal modeling capabilities of large language models, offering strong scalability and generalizability, as demonstrated by systems such as LlamaGen (Sun et al., 2024) and Open-MAGVIT2 (Luo et al., 2024). Within this family, visual autoregressive (VAR) (Tian et al., 2024) models have recently emerged as a promising direction. Unlike conventional AR models that predict the *next token* in sequence, VAR adopts a *next-scale* prediction paradigm, allowing the model to capture hierarchical visual structures and improving both quality and efficiency. Representative works, including Infinity (Han et al., 2025), HART (Tang et al., 2025), Switti (Voronov et al., 2024), and Star-T2I Ma et al. (2024), demonstrate that VAR can achieve high-resolution and high-quality T2I synthesis. Nevertheless, controllable generation with VAR remains largely unexplored. Existing methods such as ControlVAR (Li et al., 2024c) and CAR (Yao et al., 2024) are limited to class-conditioned generation on ImageNet, fail to demonstrate effectiveness in high-quality T2I settings, and rely on resource-intensive fine-tuning, which restricts their scalability. These limitations pose a key challenge–how to efficiently and effectively inject conditional information without disrupting base generation?

In this paper, to address the aforementioned challenge, we propose **ScaleWeaver**, a novel framework for efficient and effective controllable T2I generation built on VAR models. In ScaleWeaver, conditional inputs (*e.g.*, canny edges or depth maps) are tokenized using the same tokenizer as the input image, which is proven to be effective. These conditional tokens are then processed through a dedicated conditional branch, which is trained with LoRA (Hu et al., 2022) modules across all scales. The resulting conditional tokens then interact with image tokens through our proposed core module–**Reference Attention**. Reference Attention is an enhanced attention mechanism integrated into the MMDiT block (Esser et al., 2024), designed to incorporate control information with high flexibility and effectiveness. In contrast to original MM attention, Reference Attention removes the attention from image→condition that is computationally redundant for condition injection. Besides, we apply a **parameter reuse** strategy with additional linear projectors to exploit the autoregressive backbone's existing capacity to process conditional information. This greatly reduces training overhead while improving adaptation efficiency, making the method both practical and scalable for large models. The overall design of Reference Attention allows the model to progressively learn meaningful control while preserving the generative quality of the base model.

Extensive experiments demonstrate that ScaleWeaver achieves high-quality generation and robust text-image alignment across controllable generation tasks with a wide range of conditions. Our approach consistently maintains strong performance and adaptability across various types of control signals. Notably, ScaleWeaver offers a substantial improvement in efficiency over state-of-the-art diffusion-based control methods, making it a practical and scalable solution for controllable T2I generation in the autoregressive paradigm.

Our main contributions can be summarized as follows:

- We propose **ScaleWeaver**, a novel framework for controllable generation based on text-to-image VAR models, equipped with light-weight condition injection, inheriting the inference advantage and scale-wise bidirectional modeling.
- We propose **Reference Attention** mechanism for stable multi-scale integration of the condition with a significant reduction of computational cost compared with widely-used MM-Attention. Cooperating with a **parameter reuse** and zero-init strategy, Reference Attention further reduces training cost while maintaining strong adaptability.
- Extensive qualitative and quantitative evaluations show that our ScaleWeaver achieves superior controllability and efficiency compared to existing diffusion-based controllable generation methods.

## 2 RELATED WORKS

### 2.1 AUTOREGRESSIVE MODELS IN VISUAL GENERATION

Autoregressive (AR) models have long been applied to visual generation by modeling images as sequences of discrete tokens. Early works such as VQVAE (van den Oord et al., 2017) and VQ-GAN (Esser et al., 2021) tokenize images into codebook indices and employ transformer-based decoders to autoregressively generate visual content. More recent advances leverage large-scale language modeling techniques for image synthesis, as seen in LlamaGen (Sun et al., 2024) and Open-MAGVIT2 (Luo et al., 2024), which utilize powerful transformer backbones to scale up AR generation. In parallel, MAR (Li et al., 2024b) and NOVA (Deng et al., 2024) remove vector quantization and directly model continuous latents with diffusion loss, improving fidelity and scalability for images and videos. Within AR, *visual autoregressive (VAR)* (Tian et al., 2024) approaches reconceptualize the autoregressive modeling by adopting *next-scale* (coarse-to-fine) prediction rather than next-token prediction, preserving hierarchical structure and enabling scalable autoregressive image synthesis. Recent systems such as Infinity (Han et al., 2025), HART (Tang et al., 2025), Switti (Voronov et al., 2024), and Star-T2I (Ma et al., 2024) further demonstrate that VAR-based text-to-image models can achieve performance comparable to state-of-the-art diffusion models, leveraging scale-wise transformers and coarse-to-fine generation to enable high-resolution synthesis with strong text alignment. These works establish VAR as a competitive backbone for high-quality T2I generation. Besides image generation, VAR-style models extend naturally to other pixel-to-pixel vision tasks, including super-resolution (VARSR) (Qu et al., 2025), image restoration (Varformer) (Wang et al., 2025), and unified generation frameworks (VARGPT) (Zhuang et al., 2025). These results underscore VAR's versatility and scalability across visual generation tasks.

### 2.2 CONTROLLABLE IMAGE GENERATION

Controllable image generation methods aim to enable fine-grained control by injecting external conditional information into the synthesis process. Adapter-based approaches, notably ControlNet and T2I-Adapter, attach condition encoders and lightweight modulation heads to pretrained diffusion backbones, conditioning on edges, depth, normals, or segmentation while largely freezing the backbone (Zhang et al., 2023; Mou et al., 2024). Subsequent frameworks such as UniControl (Qin et al., 2023), Uni-ControlNet (Zhao et al., 2023), and ControlNet++ (Li et al., 2024a) broaden the condition space and optimized training strategies. In parallel, attention-based schemes like OmniControl (Tan et al., 2024) and EasyControl (Zhang et al., 2025) leverage multimodal attention to integrate conditions within the denoising transformer, enhancing spatial alignment and flexibility. With the advent of autoregressive (AR) backbones, research on controllable generation has increasingly extended to AR models. ControlAR (Li et al., 2025) introduces conditional decoding for Llama-Gen (Sun et al., 2024), enabling precise control in an AR setting. In the context of visual autoregressive models, ControlVAR (Li et al., 2024c), CAR (Yao et al., 2024), and SCALAR (Xu et al., 2025) incorporate conditioning into scale-wise generation, demonstrating controllability within the VAR framework. However, these methods are restricted to class-conditional generation; high-quality image generation with text prompts has not been well explored.

## 3 METHOD

### 3.1 PRELIMINARY OF VISUAL AUTOREGRESSIVE GENERATION

Different from conventional autoregresstive model based on raster-scan *next-token* decoding, Visual autoregressive (VAR) model is a innovative *next-scale* prediction framework. By conditioning each finer scale on coarser context and text prompt, VAR aligns with a coarse-to-fine perceptual progression while satisfying the autoregressive premise on newly defined causal units. Essentially, the autoregressive unit is an entire token map at a given scale rather than a single token, which reduces the number of autoregressive rounds while retains *bidirectional correlations* within each scale.

VAR model incorporates a multi-scale visual tokenizer and a generative transformer for image synthesis. The tokenizer encodes an image $\mathbf{I}$ into $K$ residual maps $\{\mathbf{R}_1, \ldots, \mathbf{R}_K\}$ from coarse to fine, and a text encoder transforms the text prompt into embedding $\mathbf{t}$. Then the generative transformer is trained to predict the next scale image map $\mathbf{R}_s$ conditioned on previous scales' maps and text

embedding, which factorizes the joint probability distribution into the conditional distributions:

$$p_\theta(\mathbf{R}_{1:K}) = \prod_{s=1}^{K} p_\theta(\mathbf{R}s \mid \mathbf{R}_{<s}, \mathbf{t}), \tag{1}$$

where $\theta$ is the weight of the generative transformer. During training, VAR model applies teacher forcing to supply ground-truth coarse scales; at inference, the model samples sequentially from coarse to fine and detokenizes the residual hierarchy to get the final image.

## 3.2 OVERVIEW OF SCALEWEAVER

To support a condition image input, we extend the above vanilla VAR architecture to incorporate an auxiliary visual condition $\mathbf{c}$. To be specific, the multi-scale tokenizer is first used to map the conditional input to multi-scale tokens $\{\mathbf{C}_1, \ldots, \mathbf{C}_K\}$. Then the generator predicts the next-scale image tokens conditioned on image tokens and the condition tokens of coarser scales, and text prompt, which is formulated as follows:

$$p_\theta(\mathbf{R}_{1:K} \mid \mathbf{t}, \mathbf{c}) = \prod_{s=1}^{K} p_\theta(\mathbf{R}_s \mid \mathbf{R}_{<s}, \mathbf{t}, \mathbf{C}_{<s}), \tag{2}$$

where condition tokens serve as side inputs and are not involved in autoregressively output.

**Overall design.** As illustrated in Fig. 2 (a), ScaleWeaver employs the same multi-scale tokenizer for both image and condition, ensuring spatial and scale alignment between $\mathbf{R}_s$ and $\mathbf{C}_s$. Conditional tokens are processed by a conditional branch that mirrors the backbone interfaces and is applied at every scale. During next-scale prediction, image tokens interact with condition tokens via the proposed *Reference Attention* module inserted in the attention block of an MMDiT-style transformer; this fusion occurs at each scale while the rest of the transformer structure remains unchanged.

**Efficient parameter reuse with LoRA.** To reduce training cost as much as possible, ScaleWeaver reuses text-to-image backbone weights and introduces LoRA adapters in the conditional branch projections across scales. LoRA modules (with small rank $r$) modulate the condition-side projections and are applied to projection layers with the condition embedder; image-side weights are frozen to preserve base model's generative capability. This strategy focuses on inheriting capacity on condition processing, keeps the number of additional parameters minimal, and enables efficient training.

**Multi-modal classifier-free guidance.** We adopt standard VAR training with a multi-scale cross-entropy objective on image tokens only; conditional tokens are auxiliary inputs and receive no direct supervision. Besides the original classifier-free guidance (CFG) used in VAR models, to enable CFG with conditional information, we randomly drop the condition during training (replace the condition image with a blank input) with probability $p$, thereby exposing the model to both conditional and non-conditional regimes. At inference, guidance operates on the logits over the next-scale token vocabulary at each scale $s$. Let $\mathbf{z}_s(\cdot)$ denote the pre-softmax logits predicted under a particular conditioning setup. The final multi-modal CFG is computed by:

$$\mathbf{z}_s^{\text{guid}} = \mathbf{z}_s^{\text{base}} + \gamma_{\text{img}}\big(\mathbf{z}_s^{\text{img}} - \mathbf{z}_s^{\text{base}}\big) + \gamma\big(\mathbf{z}_s^{\text{both}} - \mathbf{z}_s^{\text{img}}\big), \tag{3}$$

where $\mathbf{z}_s^{\text{base}}$ is the prediction based on neither text nor image, $\mathbf{z}_s^{\text{img}}$ receives only the image condition, and $\mathbf{z}_s^{\text{both}}$ receives both text and image; $\gamma_{\text{img}}, \gamma \geq 0$ are hyperparameters controlling image- and text-guided strengths, respectively. The final output is obtained by $\text{Softmax}(\mathbf{z}_s^{\text{guid}})$ at each scale.

## 3.3 REFERENCE ATTENTION

The core module in our ScaleWeaver is Reference Attention, which targets efficiently and effectively injecting the control image. We incorporate attention-based fusion of conditional and image tokens, using a mechanism designed to balance flexibility with stability. Unlike OmniControl (Tan et al., 2024) and EasyControl (Zhang et al., 2025), which employ standard multi-modal attention (Esser et al., 2024) for condition injection, the Reference Attention in our approach achieves zero-injection

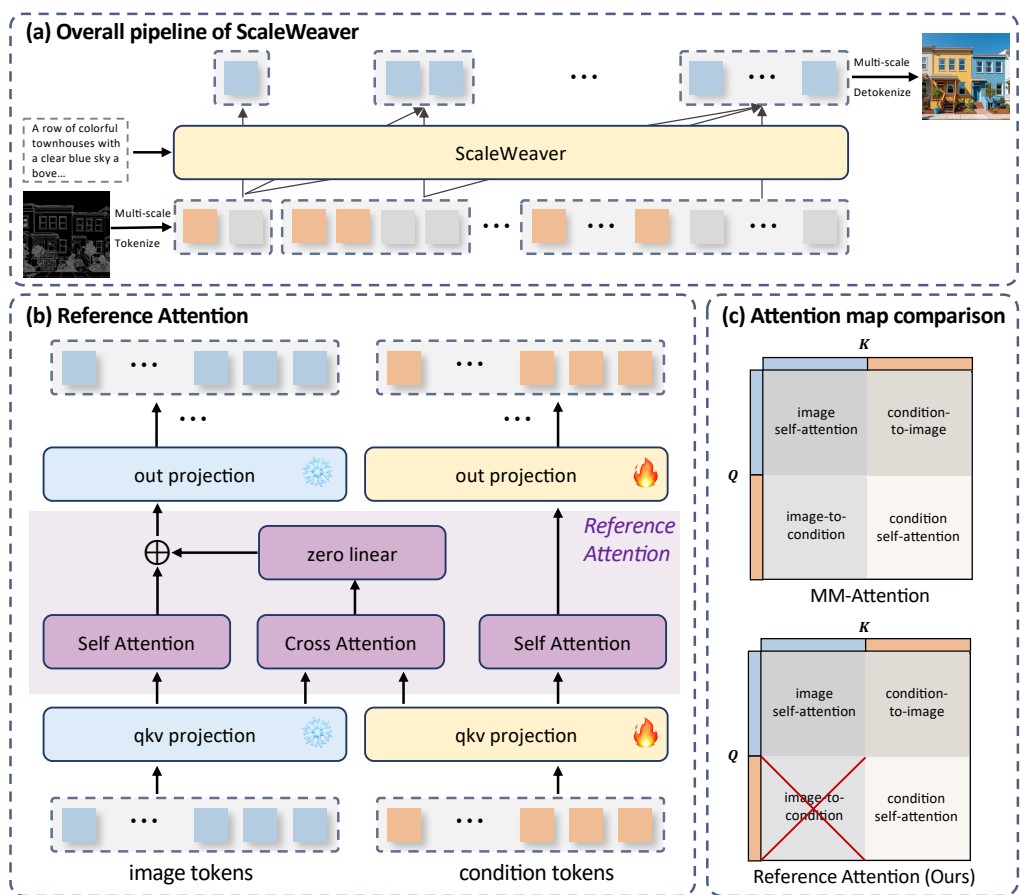

Figure 2: **Illustration of ScaleWeaver framework.** The conditional input is tokenized using the same multi-scale tokenizer as the image, processed by a LoRA-based conditional branch, and fused with image tokens via *Reference Attention* at each scale. In the Reference Attention module, image-side projections are frozen while the conditional branch is LoRA-tuned. Image queries attend to condition keys/values through cross-attention, gated by a zero-initialized projection to preserve base generation capability and enable gradual control. Unlike MM-Attention, Reference Attention removes the image→condition path, reducing unnecessary computation and stabilizing control injection. For clarity, image-text cross-attention and FFN within the original Infinity blocks are omitted.

at initialization, ensuring that conditional information is introduced without disrupting the base generation process. This design choice allows control to emerge progressively during training while maintaining the autoregressive backbone's generative prior.

What's more, Multi-modal attention typically admits four routes: image/condition self-attention and bi-directional cross-attention. Reference Attention explicitly removes the image→condition path, retaining condition→image and both self-attentions. This asymmetry reduces representational entanglement, simplifies optimization, and concentrates parameter updates on the conditional branch, which we found important for stable controllability without degrading image priors.

Let $\mathbf{X}_s^{(i)} \in \mathbb{R}^{L_s \times d}$ denote image tokens and $\mathbf{C}_s^{(i)} \in \mathbb{R}^{L_s^c \times d}$ denote condition tokens at scale $s$. We perform self-attention independently on each stream and enable a single cross path from condition to image. We derive query, key, and value projections separately for the two pathways:

$$
\begin{aligned}
[Q_x, K_x, V_x] &= \mathbf{X}_s^{(i)} \left[ W_Q, W_K, W_V \right], \\
[Q_c, K_c, V_c] &= \mathbf{C}_s^{(i)} \left[ W_Q + \Delta W_Q, W_K + \Delta W_K, W_V + \Delta W_V \right].
\end{aligned}
\tag{4}
$$

This setup highlights that the image pathway maintains frozen backbone projections, while the condition pathway learns lightweight low-rank updates to encode control information. With these pro-

jections, Reference Attention updates tokens as follows:

$$\hat{\mathbf{X}}_s^{(i)} = \text{Attn}(Q_x, K_x, V_x), \ \hat{\mathbf{C}}s^{(i)} = \text{Attn}(Q_c, K_c, V_c),$$
$$\widetilde{\mathbf{X}}_s^{(i)} = \hat{\mathbf{X}}_s^{(i)} + W_{\text{zero}}\text{Attn}(Q_x, K_c, V_c), \tag{5}$$

where $W_{\text{zero}}$ is zero-initialized, ensuring that at start the module behaves identically to standard self-attention, while gradually learning to incorporate conditional guidance as training progresses.

Reference Attention acts as a drop-in replacement inside the attention block of an MMDiT-style transformer. At each scale $s$, the module receives $\mathbf{X}_s^{(i)}$ and $\mathbf{C}_s^{(i)}$, applies stream-wise self-attention, then injects zero-initiated condition via Eq. 5. The module is applied at all scales, enabling coarse guidance at low $s$ and refinement at high $s$, while other layers remain unchanged. Since image-side projections are frozen and condition-side projections are LoRA-tuned with a small rank, the number of trainable parameters is minimal, maintaining computational efficiency.

## 4 EXPERIMENTS

### 4.1 EXPERIMENTAL SETUP

**Tasks and conditions.** We conduct experiments on six conditional generation tasks that span diverse structural and appearance-based controls: Canny edge, depth (Yang et al., 2024), blur, colorization, color palette, and sketch (Su et al., 2021) images. These tasks are chosen to cover both geometric guidance and appearance or style-related guidance, thereby providing a comprehensive assessment of controllability and image quality.

**Training details.** Training is conducted on a composite dataset of 26k high-resolution images generated with FLUX.1-dev from the text-to-image-2M dataset (Hate, 2024) and fluxdev-controlnet-16k dataset (Kadirnar, 2024). We use Infinity 2B as the autoregressive backbone and fine-tune with LoRA-Plus (Hayou et al., 2024) optimizer on 4 NVIDIA A40 GPUs. Only LoRA modules and zero-linear layers in the conditional branch are updated, while the backbone remains frozen. For more details, please refer to Appendix A.1.

**Evaluation metrics.** We assess both generation quality and controllability. For image quality, we report Fréchet Inception Distance (FID)(Heusel et al., 2017) and CLIP-IQA (Wang et al., 2023), and use CLIP Score (Radford et al., 2021) to evaluate text-image alignment. For control fidelity, we use F1 score for Canny-based edge control and mean squared error (MSE) for the remaining conditions. All evaluations are conducted on 5,000 images from the COCO 2017 (Lin et al., 2014) validation set. This evaluation protocol provides a balanced view of fidelity, alignment, and controllability.

### 4.2 EXPERIMENTAL RESULTS

**Quantitative comparison.** We compare ScaleWeaver with a comprehensive set of controllable generation baselines, including SD1.5-based ControlNet (Zhang et al., 2023) and T2I-Adapter (Mou et al., 2024), as well as FLUX.1-based ControlNet Pro, OminiControl (Tan et al., 2024), and Easy-Control (Zhang et al., 2025). ScaleWeaver is evaluated across diverse condition types, demonstrating effective and precise control in all settings. Notably, for challenging conditions such as depth and blur, our controllability surpasses existing methods. Across all tasks, ScaleWeaver maintains strong generative quality and text-image consistency, achieving results on par with leading diffusion-based approaches. Importantly, these advances are realized with significantly improved efficiency, underscoring the practical advantages of our approach for controllable generation.

**Qualitative comparison.** As shown in Fig. 3, our method consistently produces images with clear, well-defined subjects and distinct boundaries, while maintaining both foreground and background integrity. The generated images exhibit high visual coherence, with minimal artifacts and strong separation between objects and their surroundings. Furthermore, our approach demonstrates robust text-image alignment: even for prompts with long or complex descriptions, ScaleWeaver accurately captures fine-grained details and generates images that faithfully reflect the input text.

| Condition | Method | Base model | Controllability F1 ↑ /MSE ↓ | Generative Quality FID ↓ | CLIP-IQA ↑ | Text Consistency CLIP-Score ↑ |
|---|---|---|---|---|---|---|
| Depth | ControlNet | SD 1.5 | 923 | **23.03** | 0.64 | 0.308 |
| | T2I-Adapter | | 1560 | 24.72 | 0.61 | **0.309** |
| | ControlNet Pro | | 2958 | 62.20 | 0.55 | 0.212 |
| | OminiControl† | FLUX.1-dev | 556 | 30.75 | **0.68** | 0.307 |
| | EasyControl† | | 607 | 23.04 | 0.57 | 0.303 |
| | Ours | Infinity 2B | **506** | 25.80 | 0.67 | 0.302 |
| Canny | ControlNet | SD 1.5 | 0.35 | 18.74 | 0.65 | 0.305 |
| | T2I-Adapter | | 0.22 | 20.06 | 0.57 | 0.305 |
| | ControlNet Pro | | 0.21 | 98.69 | 0.48 | 0.192 |
| | OminiControl† | FLUX.1-dev | **0.45** | 23.63 | 0.66 | **0.306** |
| | EasyControl† | | 0.32 | **18.47** | 0.60 | 0.303 |
| | Ours | Infinity 2B | 0.30 | 22.31 | **0.66** | 0.299 |
| Colorization | ControlNet Pro | FLUX.1-dev | 994 | 30.38 | 0.40 | 0.279 |
| | OminiControl† | | **109** | 9.76 | **0.56** | **0.312** |
| | Ours | Infinity 2B | 186 | **9.34** | 0.48 | 0.310 |
| Deblur | ControlNet Pro | FLUX.1-dev | 338 | **16.27** | 0.55 | 0.294 |
| | OminiControl | | 62 | 18.89 | **0.59** | 0.301 |
| | Ours | Infinity 2B | **46** | 17.39 | 0.53 | **0.308** |

Table 1: **Quantitative comparison** of controllable T2I generation on COCO 2017. Best results are shown in **bold** and second best results are underlined. † indicates that the results are reproduced under the same testing protocol to ensure a fair comparison.

| Method | Generation Quality ↑ | Controllability ↑ | Text-Image Alignment ↑ | Overall ↑ |
|---|---|---|---|---|
| OminiControl | 0.3053 | **0.3474** | 0.2947 | 0.3158 |
| ControlNet | 0.0702 | 0.1053 | 0.0912 | 0.0889 |
| EasyControl | 0.1719 | 0.2702 | 0.2351 | 0.2257 |
| Ours | **0.4526** | 0.2772 | **0.3789** | **0.3696** |

Table 2: **Human evaluation** in terms of generation quality, controllability, text-image alignment. Each value indicates the percentage of times a method was preferred by users (higher is better). ScaleWeaver outperforms all baselines across all criteria.

**Diverse generation.** As demonstrated in Fig. 4, ScaleWeaver is capable of generating images with diverse styles and content for different user prompts while preserving a fixed semantic structure provided by the condition. This highlights the effectiveness of our controllable generation framework in delivering both high fidelity and precise semantic correspondence, as well as supporting diverse user intent under consistent structural guidance.

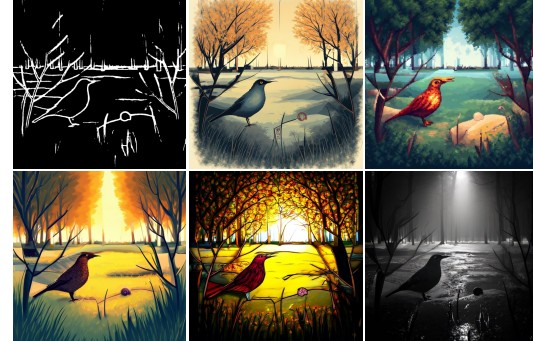

**Human Evaluation.** To further validate the effectiveness of our approach, we conduct a user study in which participants are asked to select the best image according to three criteria: image quality, controllability and text-image alignment. The results of the user study are consistent with our quantitative findings, confirming that ScaleWeaver achieves competitive

Figure 4: **Diverse generation** with the same sketch condition but different text prompts.

controllability and high-quality text-to-image generation compared to baseline methods.

**Analysis of Efficiency.** We compare ScaleWeaver's efficiency with state-of-the-art diffusion-based methods in Tab. 3. All experiments are conducted on $1024 \times 1024$ image generation. For

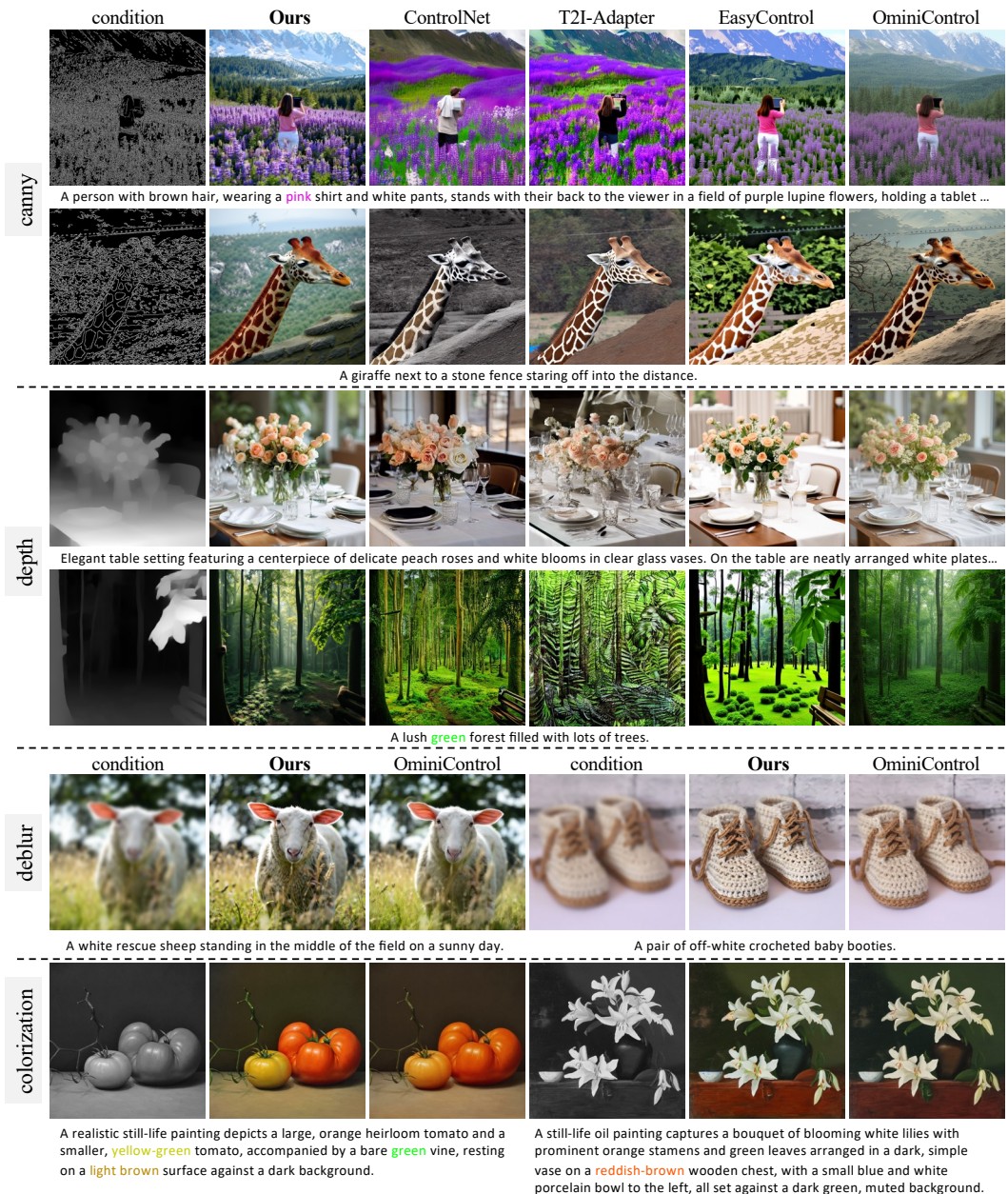

Figure 3: **Qualitative comparison** of controllable text-to-image generation results. ScaleWeaver achieves high-fidelity synthesis and precise control across diverse conditions, outperforming diffusion-based baselines in both visual quality and controllability. Best viewed via zoom in.

diffusion-based baselines, we use 28 sampling steps, following standard practice. Inference latency is measured as the wall-clock time required to generate an image on an NVIDIA A40 GPU.

Our method achieves comparable or better generation quality with a significantly smaller model size (2.3B parameters) compared to FLUX.1 (Labs, 2024) based methods. More importantly, ScaleWeaver delivers a substantial speedup in inference, requiring only 7.6 seconds per image, which is **5–9×** **faster** than FLUX.1-based approaches. This demonstrates the efficacy of our approach and highlights the practical advantage of controllable

| Method | #Param (M) | Latency (s) |
|---|---|---|
| ControlNet Pro | 15B | 38.7 |
| OminiControl | 12B | 70.9 |
| EasyControl | 12B | 60.4 |
| Ours | **2.3B** | **7.6** |

Table 3: **Efficiency comparison** for $1024 \times 1024$ image generation.

generation with VAR-based models, making ScaleWeaver a highly efficient and scalable solution for real-world applications.

### 4.3 ABLATION STUDY

To better understand which injection mechanisms are most effective for spatial control in visual autoregressive (VAR) text-to-image models, we conducted ablation studies on injection mechanisms and conditional branch configurations. All ablation experiments are conducted on the Canny condition, with each model trained for 24k iterations. During the evaluation, guidance is applied solely to the condition image with a guidance scale of 1.5.

| Injection Method | F1 ↑ | FID ↓ | CLIP Score ↑ |
|---|---|---|---|
| Spatial addition | 0.315 | 25.448 | 0.296 |
| MM-Attention | **0.344** | 24.888 | 0.294 |
| Ours | 0.325 | **23.224** | **0.298** |

Table 4: **Ablation study on injection methods.**

**Injection mechanisms.** We compare three injection mechanisms: spatial addition, multi-modal attention (MM-Attention), and our proposed Reference Attention. As shown in Tab. 4, MM-Attention achieves the highest controllability (F1 score) but at the cost of degraded generation quality and text-image alignment. Our Reference Attention achieves the best overall balance, delivering strong controllability while preserving high generation quality and text alignment. Please refer to Appendix A.2 for visual comparison. This demonstrates that Reference Attention effectively integrates conditional information without disrupting the base generative prior.

**Design choices.** We perform comprehensive ablation studies to investigate the key components in our ScaleWeaver, including zero-initialized linear gating, LoRA rank, condition injection location, and number of conditional blocks. As shown in Tab. 5, the default setting (zero-linear enabled, LoRA rank 16, condition injected in the first 16 blocks) achieves the best balance of controllability and generation quality. Removing the zero-linear gate signif-

| Component | Setting | F1 ↑ | FID ↓ | CLIP Score ↑ |
|---|---|---|---|---|
| zero-linear | w/ → w/o | 0.282 | 25.059 | 0.296 |
| LoRA Rank | 16 → 4 | 0.320 | 25.703 | 0.296 |
| | 16 → 8 | 0.322 | 25.712 | 0.296 |
| Condition Blocks | first 16 → last 16 | 0.344 | 24.510 | 0.296 |
| | first 16 → all | **0.346** | 26.211 | 0.296 |
| Number of Blocks | 16 → 4 | 0.202 | 25.367 | **0.299** |
| | 16 → 8 | 0.229 | 24.132 | 0.298 |
| Default | Default | 0.325 | **23.224** | 0.298 |

Table 5: **Ablation studies on key design choices.**

icantly reduces controllability, highlighting its importance for stable control injection. Lowering the LoRA rank from 16 to 4 or 8 slightly decreases controllability and worsens FID. While injecting conditions in the last 16 blocks or all blocks increases controllability, it suffers considerable degradation in image quality, suggesting that early integration is more effective. Finally, reducing the number of conditional blocks from 16 to 4 or 8 degrades controllability, underscoring the need for sufficient capacity to process conditional information. Overall, these ablations confirm that our design choices in ScaleWeaver achieve the best trade-off between controllability and generation quality.

## 5 CONCLUSION

In this paper, we introduced ScaleWeaver, a parameter-efficient and scalable framework for controllable text-to-image generation based on visual autoregressive models. By leveraging a novel Reference Attention mechanism and a parameter reuse strategy with LoRA, ScaleWeaver enables precise and flexible multi-scale control while maintaining high generation quality and efficiency. Extensive experiments demonstrate that our approach achieves competitive controllability and faster inference compared to diffusion-based baselines, with strong text-image alignment and visual fidelity. We believe ScaleWeaver provides a practical and extensible foundation for future research on efficient and controllable generative modeling in the autoregressive paradigm.

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

# A APPENDIX

## A.1 MORE IMPLEMENTATION DETAILS

We train all models using 4 NVIDIA A40 GPUs, with a per-GPU batch size of 2. Training is performed using the LoRA-Plus optimizer with a learning rate of 5e-5 and a LoRA rank of 16. Only the LoRA modules and zero-linear layers in the conditional branch are updated; the image/text backbone remains frozen. The default training hyperparameters are summarized in Table 6.

| Hyperparameter | Default Setting |
|---|:---:|
| Gradient clipping | 5 |
| Adam betas | (0.9, 0.97) |
| LoRAPlus LR ratio | 1.25 |
| Batch size | 8 |
| Learning rate | 5e-5 |
| Learning rate decay | None |
| Mixed precision | bfloat16 |
| Reweight loss by scale | True |
| Zero-linear gating | Enabled |
| LoRA rank | 16 |
| Condition injection location | First 16 blocks |

Table 6: Training config.

Table 7 summarizes the number of training iterations for each condition type. All models converge efficiently, typically within 30k iterations, demonstrating the practicality of our approach.

| Condition Type | Training Iterations |
|---|:---:|
| Canny Edge | 36,000 |
| Depth Map | 24,000 |
| Colorization | 24,000 |
| Deblur | 24,000 |
| Sketch | 24,000 |
| Palette | 24,000 |

Table 7: Training iterations for each condition type.

## A.2 VISUALIZATION COMPARISON FOR ABLATION STUDY

In this section, we provide qualitative comparison of different injection mechanisms evaluated in our ablation study, as illustrated in Figure 5. By zooming in on the generated images, it is evident that our proposed injection method produces more realistic subjects with fewer artifacts compared to alternative approaches. This demonstrates the effectiveness of our injection strategy in integrating conditional information while maintaining image quality.

## A.3 MORE VISUALIZATION RESULTS

Diverse generation with the same sketch condition is shown in Fig. 6 along with the prompts we used. More visualization results under different control conditions are shown in Figs. 7, 8, 9. These examples further demonstrate that our approach is capable of faithfully following diverse types of control signals, while simultaneously producing outputs with rich variations in appearance and structure. The results indicate that our method not only achieves precise controllability but also preserves a high degree of generative diversity, which is crucial for adapting to different application scenarios.

## A.4 LIMITATION AND FUTURE WORK

**Limitation.** While ScaleWeaver demonstrates strong performance across a range of conditions, there are some limitations to consider. First, the generation capability is ultimately determined by the

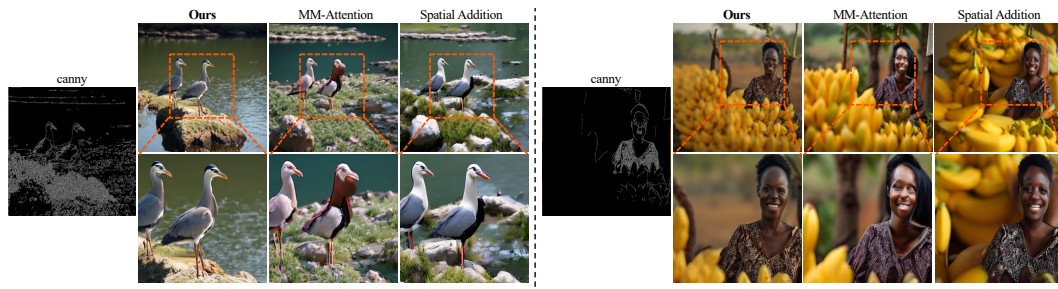

Figure 5: Visualization comparison for ablation study. We show qualitative results for different injection mechanisms.

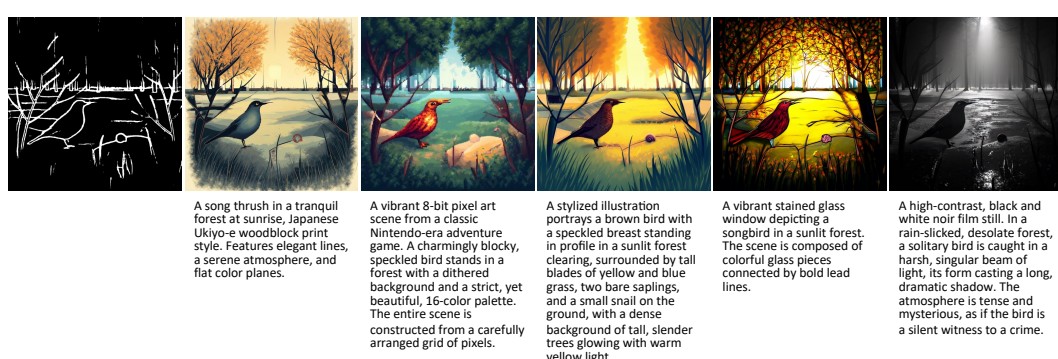

Figure 6: Conditional generation results on diverse condition types.

capacity of the underlying base model; with our efficient training strategy, we believe ScaleWeaver can perform well and further benefit from scaling to larger VAR models. Second, our method does not currently support multi-condition control simultaneously.

**Future work.** Future research directions include extending ScaleWeaver to support multi-condition control, as the current framework is limited to handling a single condition at a time. The Reference Attention mechanism, with its modular design, naturally supports multi-condition integration, making this a promising avenue for exploration. Additionally, investigating methods to dynamically balance the influence of multiple conditions could further enhance the flexibility and adaptability of the model. Another potential direction is to scale ScaleWeaver to larger VAR backbones, leveraging the scalability of autoregressive models to improve generation quality and controllability.

## A.5 THE USE OF LARGE LANGUAGE MODELS (LLMS)

We utilize a large language model to assist in correcting grammar errors and polishing the language of this paper. All ideas, methods, experiments, and conclusions are the sole work of the authors.

756
757
758
759
760
761
762
763
764
765
766
767
768
769
770
771
772
773
774
775
776
777
778
779
780
781
782
783
784
785
786
787
788
789
790
791
792
793
794
795
796
797
798
799
800
801
802
803
804
805
806
807
808
809

Condition          Generated Images

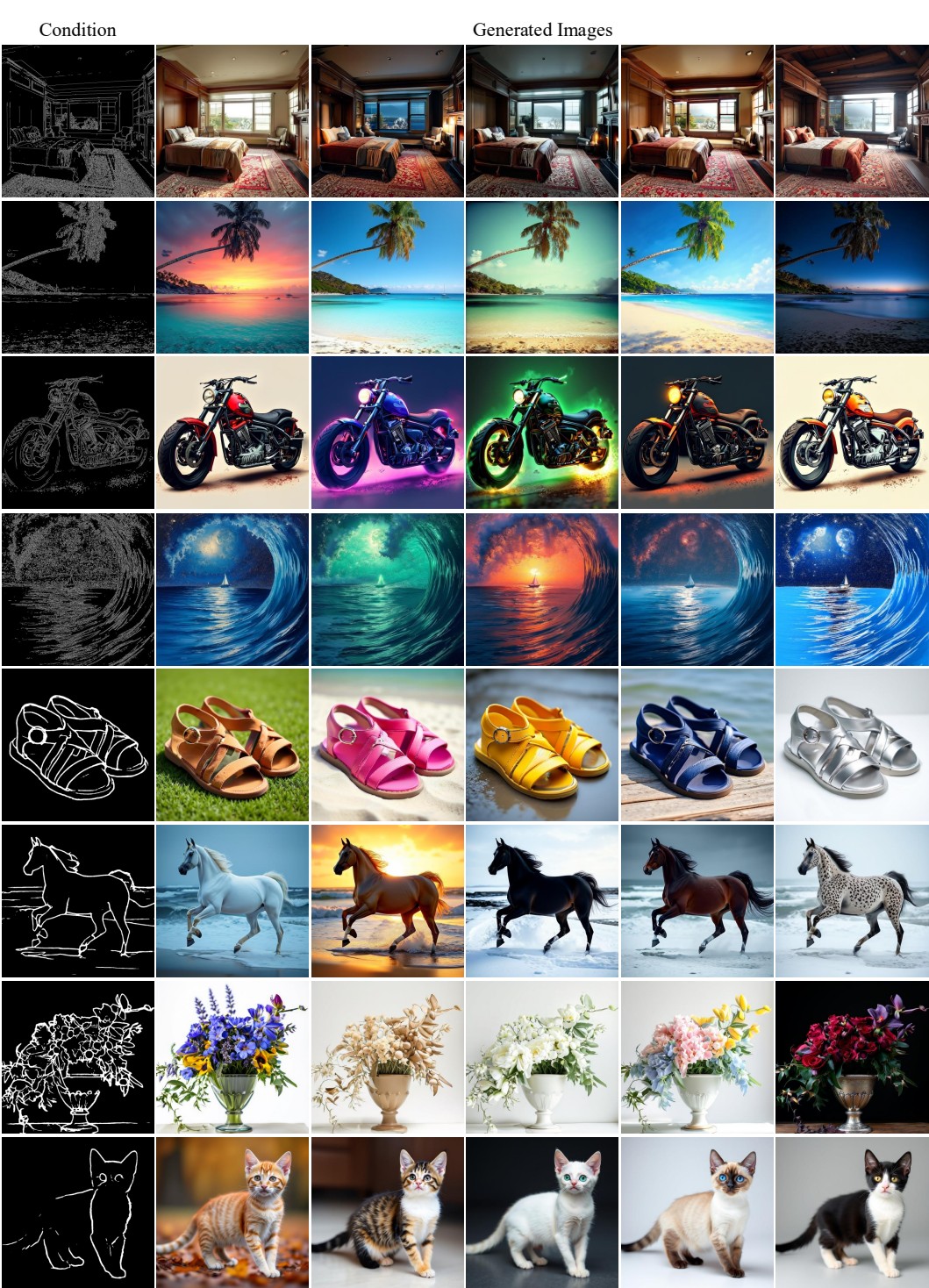

Figure 7: Diverse generation by ScaleWeaver, with Canny image or sketch image as control image.

Condition                            Generated Images

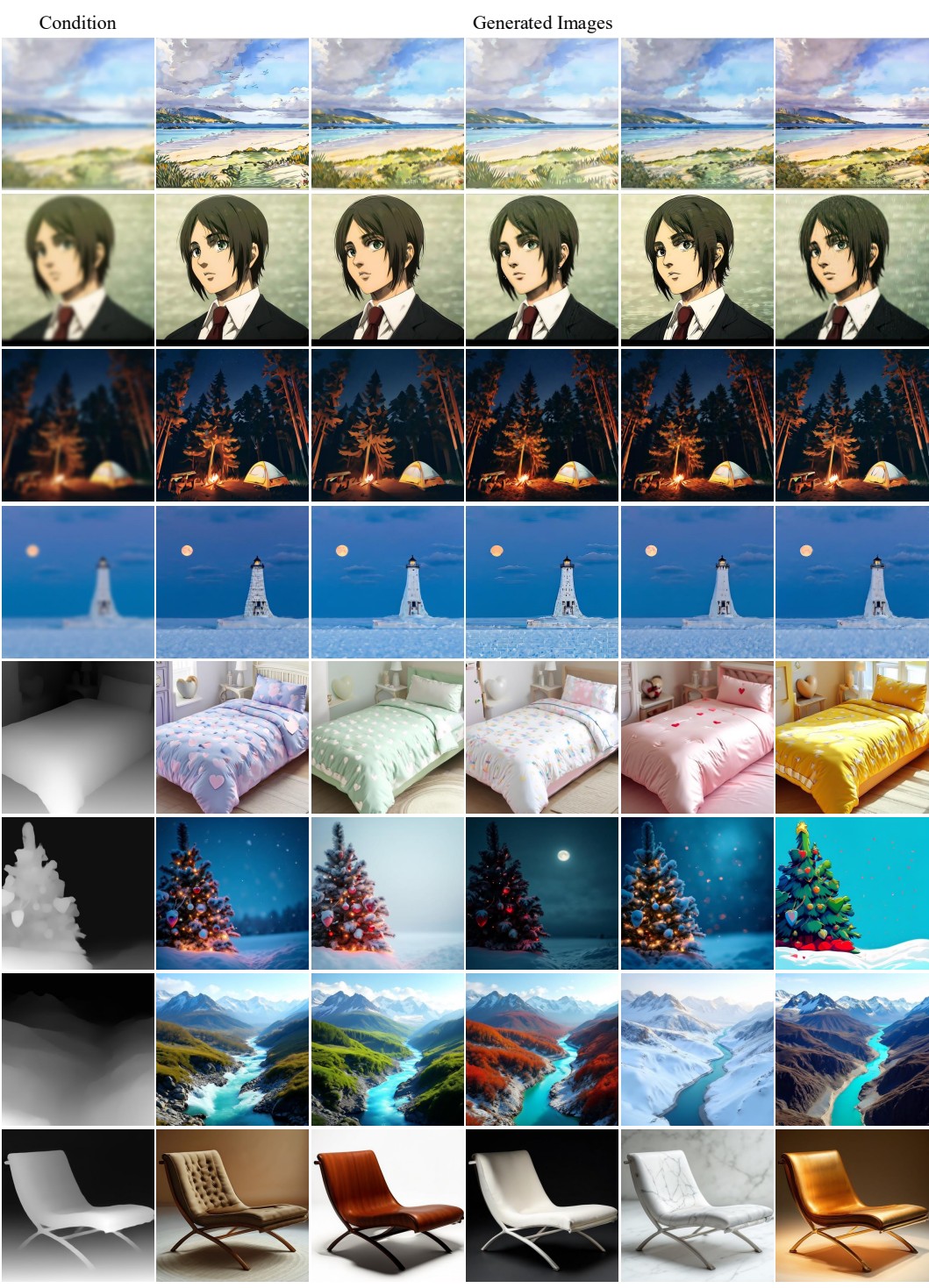

Figure 8: Diverse generation by ScaleWeaver, with blur image or depth map as control image.

Condition                    Generated Images

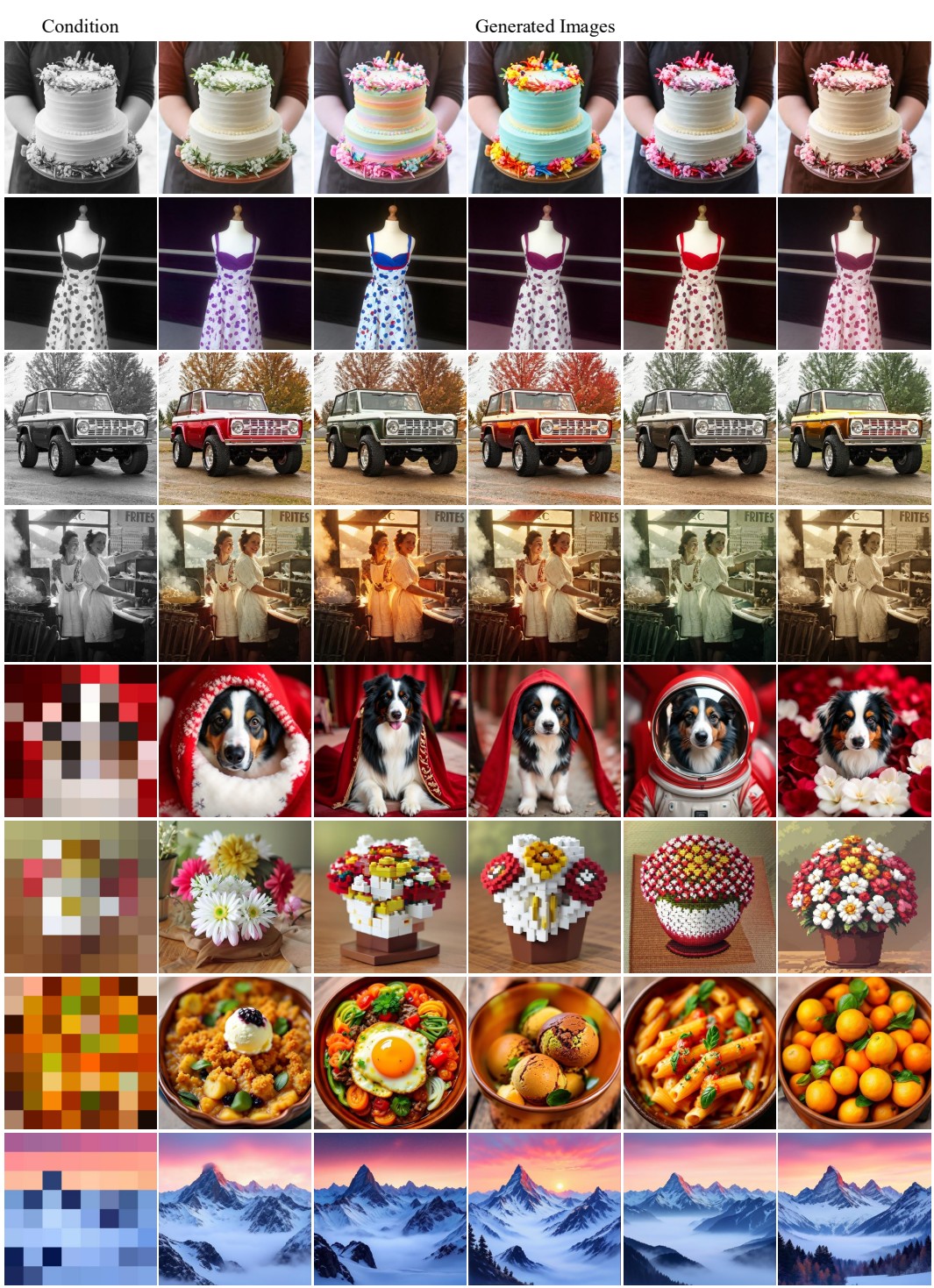

Figure 9: Diverse generation by ScaleWeaver, with grey image or palette map as control image.

