# OpenReview forum: "ScaleWeaver: Weaving Efficient Controllable T2I Generation with Multi-Scale Reference Attention"
_ICLR.cc/2026/Conference — ICLR 2026 Conference Withdrawn Submission_

### Official Review · Reviewer_N9Tq · 2025-10-31

**Soundness:** 3
**Presentation:** 3
**Contribution:** 3
**Rating:** 4
**Confidence:** 4

**Summary:**

ScaleWeaver is a framework for controllable text-to-image generation built on visual autoregressive models. It introduces a parameter-efficient fine-tuning approach to incorporate spatial control signals (e.g., edges, depth, sketches) while preserving the generative quality and inference speed of VAR backbones.

**Strengths:**

1.The paper introduces Reference Attention, a structurally simplified attention mechanism that removes redundant computation paths. This is combined with a zero-init gating and LoRA-based parameter reuse, enabling highly efficient and stable control injection.

2.The method is rigorously tested across diverse control tasks, showing strong performance against diffusion-based baselines. Comprehensive ablations definitively justify each core design choice, proving a superior trade-off between controllability and quality.

3.The framework achieves a dramatic 5-9x inference speedup over leading methods while using a smaller model, establishing VAR models as a highly practical and scalable solution for real-time controllable generation.

**Weaknesses:**

1.The efficiency claims are not sufficiently substantiated. The comparison in Table 3 conflates the gains from the efficient VAR backbone with those from the proposed Reference Attention. To isolate the contribution of the  attention mechanism, a controlled ablation is required. The Author should compare the inference latency or FLOPs of your Reference Attention against a MM-Attention baseline within the same Infinity backbone. This will directly quantify the efficiency advantage of the core contribution.

2.The ablation study on injection methods shows minimal numerical improvements, making it difficult to determine whether the performance gains stem from the base model or the proposed method. This is particularly true given the absence of controlled experiments using the same Infinity-based VAR backbone as a baseline. The authors should either conduct proper controlled comparisons or provide a compelling explanation for why ScaleWeaver achieves significant overall improvements despite only marginal differences in the ablation study.

3.The paper's choice to inject conditions into the "first 16 blocks" of the transformer, while shown to be effective through ablation studies, remains a heuristic choice. This empirically driven approach lacks deeper theoretical or principled analysis explaining why early-stage injection is more effective and whether this strategy represents a universally optimal configuration.

**Questions:**

1. To substantiate the efficiency claims of Reference Attention, can you provide a controlled comparison of inference latency or FLOPs between your method and an MM-Attention baseline, both integrated into the same Infinity backbone? This is crucial to isolate the computational contribution of your proposed module from the inherent efficiency of the base VAR model.

2.The performance differences in your ablation study (Table 4) are very small. Can you explain the mechanism or provide additional evidence clarifying how these marginal gains in the isolated component test translate to the significant overall improvements shown in the main results? Please address whether the primary advantage lies in efficiency rather than a large performance leap.

3.The choice of the "first 16 blocks" for condition injection appears heuristic. Can you provide a deeper, more principled analysis or hypothesis for why early-stage integration is more effective? For instance, does it allow the control signal to better guide the fundamental scene structure formed in the transformer's earlier layers?

---

### Official Review · Reviewer_Yyqb · 2025-10-31

**Soundness:** 3
**Presentation:** 3
**Contribution:** 1
**Rating:** 2
**Confidence:** 3

**Summary:**

This paper proposes ScaleWeaver, a framework for controllable image generation within Visual AutoRegressive (VAR) models. It extends the VAR paradigm by incorporating additional visual conditions such as edge, depth, and segmentation maps.

The core component Reference Attention enables efficient and stable integration of conditional information by retaining only the condition-to-image attention path and initializing with zero injection. The framework further adopts LoRA-based parameter reuse and multi-modal classifier-free guidance to achieve parameter-efficient and flexible conditioning.

Experiments demonstrate that ScaleWeaver produces text-aligned and controllable images with minimal parameter overhead and faster inference compared to diffusion-based approaches. Overall, the paper contributes an effective and efficient design that brings controllable generation to the VAR framework.

**Strengths:**

Originality.

- The paper introduces Reference Attention, a targeted architectural modification to multi-modal attention for controllable VAR models.
- By removing the image-to-condition pathway, freezing image-side projections, and applying zero-initialized, LoRA-based condition injections, the method achieves stable and progressive conditioning without disturbing the pretrained autoregressive prior.

Quality.

- The method is technically sound and mathematically well-formulated.
- It provides explicit factorization of multi-scale autoregressive generation, detailed equations for Reference Attention, and ablations analyzing gating, LoRA rank, and injection placement.

Clarity.

- The paper is clearly written and well-organized. Figures effectively illustrate the overall pipeline and the directional difference between standard cross-attention and Reference Attention.
- Mathematical notation and implementation details are easy to follow, and the supplementary material includes complete training settings for reproducibility.

Significance.

- ScaleWeaver successfully brings controllable generation to the VAR family, demonstrating good visual quality. It achieves faster inference and lower computational cost.

**Weaknesses:**

The main limitation of the paper lies in its limited contribution.

Although the overall design is reasonable and the implementation is solid, the core innovation focuses on the Reference Attention module, which is essentially a lightweight modification of standard multi-modal attention (such as those used in OmniControl or EasyControl).

It removes the image→condition pathway, adopts zero initialization, and applies LoRA updates. Structurally, it represents an engineering optimization for stability rather than a new generative modeling mechanism or theoretical innovation. While effective, this change lacks sufficient conceptual depth to support a top-tier level of methodological novelty.

In addition, other components in the paper (such as multi-modal classifier-free guidance, LoRA-based parameter reuse, and the teacher forcing training strategy) are direct adoptions or minor variations of existing techniques. These modifications improve efficiency from an engineering perspective but do not substantially extend the theoretical boundaries of the VAR framework.

The paper does not sufficiently demonstrate whether Reference Attention outperforms more general conditional fusion mechanisms (such as gated cross-attention, FiLM, or parameterized adapters), nor does it explore its transferability to other visual tasks, making the contribution appear limited to a specific implementation.

**Questions:**

I do not have major clarification questions regarding the current presentation. However, the authors may consider discussing related work: Training-Free Text-Guided Image Editing with Visual Autoregressive Model (Wang et al., 2025).

---

### Official Review · Reviewer_u9g8 · 2025-11-01

**Soundness:** 2
**Presentation:** 2
**Contribution:** 2
**Rating:** 2
**Confidence:** 5

**Summary:**

This paper presents Scale Weaver, a novel and parameter-efficient approach to extend Visual Autoregressive (VAR) models to handle diverse conditional T2I generation tasks. The authors address the current bias toward Diffusion Models for control by leveraging the known inference efficiency of the VAR paradigm.

**Strengths:**

* The implementation of parameter-efficient fine-tuning via LoRA on the conditional branch and the use of a zero-initialized gate are technically sound engineering decisions that promote training stability and efficiency. The approach effectively balances generation quality (competitive FID/CLIP-IQA) with control fidelity (competitive F1/MSE).
* The paper is generally well-written and follows a clear structure. The figures (especially Figure 2 ) clearly illustrate the architecture of the Reference Attention module and the overall pipeline. The authors are explicit about the base model (Infinity 2B) and the parameter-efficient nature of the training.

**Weaknesses:**

* The claim of addressing a "critical gap" in VAR-based controllable T2I is undermined by the lack of a direct comparison against existing VAR-control methods (e.g., ControlVAR, CAR, SCALAR). While the authors mention these are limited to ImageNet, a demonstration of why their architecture is superior for the T2I setting, or an ablation demonstrating superiority over a straightforward application of a ControlNet-like encoder within the VAR architecture, is missing. The novelty of Reference Attention over standard MM-Attention appears incremental; the ablation (Tab. 4) shows MM-Attention achieved the highest F1 score, suggesting the design trade-off for Reference Attention is marginal, not fundamentally superior.
* The significant efficiency claim (5-9x speedup) is fundamentally flawed as it compares a significantly smaller model (Scale Weaver, 2.3B parameters, Infinity 2B backbone) against much larger diffusion models (ControlNet Pro, 15B; OmniControl, 12B). The speedup is primarily a result of the smaller model size and the inherent efficiency of the VAR architecture, not solely the novel control mechanism.

**Questions:**

* The paper must include a quantitative comparison against existing controlled VAR methods (e.g., ControlVAR, CAR, SCALAR) to substantiate the claim of pioneering high-quality controllable T2I within the VAR paradigm. If those methods cannot be adapted to T2I, this inability should be explicitly demonstrated and technically justified.
* The efficiency comparison is currently unfair due to significant differences in model size. Please conduct an ablation study comparing the speed and quality of Scale Weaver when applied to two different-sized VAR backbones (e.g., Infinity 2B and a hypothetical 12B VAR model) to demonstrate the scaling properties of the control mechanism itself.

---

### Official Review · Reviewer_fca6 · 2025-11-04

**Soundness:** 3
**Presentation:** 3
**Contribution:** 2
**Rating:** 4
**Confidence:** 3

**Summary:**

This paper introduces ScaleWeaver, a framework for controllable text-to-image generation built on Visual Autoregressive (VAR) models. The core contribution is a Reference Attention mechanism that efficiently incorporates conditional information (e.g., canny edges, depth maps) into VAR-based generation through parameter-efficient fine-tuning. Unlike standard multi-modal attention, Reference Attention removes the image→condition attention path and employs zero-initialized projection with LoRA adapters to preserve the base model's generative capability while enabling precise control. Experiments across six condition types demonstrate competitive quality and controllability compared to diffusion-based methods, with significantly faster inference (5-9× speedup).

**Strengths:**

1.Addresses an underexplored area by extending controllable generation to the VAR paradigm, demonstrating practical applications beyond class-conditional ImageNet settings.
2.The Reference Attention mechanism is well-motivated, with clear justification for removing bidirectional cross-attention. The combination with LoRA and zero-initialization is effective for parameter-efficient training.
3.Comprehensive experimental evaluation across six diverse condition types with multiple strong baselines. The efficiency gains (7.6s vs 38-71s, 2.3B vs 12-15B parameters) are substantial and practically valuable.

**Weaknesses:**

Comparison Fairness: The comparison primarily uses FLUX.1-based diffusion models (12B parameters, 28 steps) against a 2.3B VAR backbone. While the efficiency advantage is real, a more balanced comparison would include smaller diffusion models (e.g., SD1.5-based methods receive less emphasis despite being in Table 1) or discuss scaling trends.
Limited Scope:Training dataset is relatively small (26k images) compared to large-scale controllable generation methods
Experimental Gaps:
--Human evaluation (Table 2) is limited in scale and lacks details on number of participants, evaluation protocol, and statistical significance testing
--Missing analysis on failure cases or condition types where the method struggles

**Questions:**

1. Controllability trade-offs: Table 1 shows your method sometimes achieves worse controllability metrics (e.g., Canny F1: 0.30 vs OmniControl's 0.45). Can you provide analysis on when and why this occurs? Is this an inherent trade-off with the VAR paradigm or other issue?
2. Scaling behavior: How does performance change with VAR model size? Have you experimented with larger backbones beyond Infinity 2B? Does the efficiency advantage persist at scale?
3. Generalization: How does the model perform on out-of-distribution conditions or real-world noisy inputs (e.g., hand-drawn sketches vs. clean synthetic edges)?

---

### Note · Authors · 2025-11-21

I have read and agree with the venue's withdrawal policy on behalf of myself and my co-authors.